# Computational modeling of multiscale collateral blood supply in a whole-brain-scale arterial network

**Tomohiro Otani**[1]*, **Nozomi Nishimura**[1], **Hiroshi Yamashita**[2], **Satoshi Ii**[3],
**Shigeki Yamada**[4,5], **Yoshiyuki Watanabe**[6], **Marie Oshima**[5], **Shigeo Wada**[1]

1 Graduate School of Engineering Science, Osaka University, Osaka, Japan, 2 Graduate School of Integrated Sciences for Life, Hiroshima University, Hiroshima, Japan, 3 Graduate School of Systems Design, Tokyo Metropolitan University, Tokyo, Japan, 4 Department of Neurosurgery, Nagoya City University Graduate School of Medical Science, Aichi, Japan, 5 Interfaculty Initiative in Information Studies, The University of Tokyo, Tokyo, Japan, 6 Department of Radiology, Shiga University of Medical Science, Shiga, Japan

* otani.tomohiro.es@osaka-u.ac.jp

**Data Availability Statement:** All data files underlying the findings reported in a submitted manuscript are available from the OSF database

## Abstract

The cerebral arterial network covering the brain cortex has multiscale anastomosis structures with sparse intermediate anastomoses ($O[10^2]$ μm in diameter) and dense pial networks ($O[10^1]$ μm in diameter). Recent studies indicate that collateral blood supply by cerebral arterial anastomoses has an essential role in the prognosis of acute ischemic stroke caused by large vessel occlusion. However, the physiological importance of these multiscale morphological properties—and especially of intermediate anastomoses—is poorly understood because of innate structural complexities. In this study, a computational model of multiscale anastomoses in whole-brain-scale cerebral arterial networks was developed and used to evaluate collateral blood supply by anastomoses during middle cerebral artery occlusion. Morphologically validated cerebral arterial networks were constructed by combining medical imaging data and mathematical modeling. Sparse intermediate anastomoses were assigned between adjacent main arterial branches; the pial arterial network was modeled as a dense network structure. Blood flow distributions in the arterial network during middle cerebral artery occlusion simulations were computed. Collateral blood supply by intermediate anastomoses increased sharply with increasing numbers of anastomoses and provided one-order-higher flow recoveries to the occluded region (15%–30%) compared with simulations using a pial network only, even with a small number of intermediate anastomoses ($\leq 10$). These findings demonstrate the importance of sparse intermediate anastomoses, which are generally considered redundant structures in cerebral infarction, and provide insights into the physiological significance of the multiscale properties of arterial anastomoses.

link: https://osf.io/fn3jb/?view_only=ab80e6c936df44d5ac8c9aad7fe12cbf.

**Funding:** This work was supported by MEXT as the "Program for Promoting Researches on the Supercomputer Fugaku" (hp220161 to SW and hp230208 to SI), the Japanese Society for the Promotion of Science for Grants-in-Aid for Scientific Research (No. 19H01175 to SW, 21K18037 to TO, 22H00190 to MO, 23K11830 to TO), and the Multidisciplinary Research Laboratory System for Future Developments, Osaka University Graduate School of Engineering Science (to TO). The funders had no role in study design, data collection and analysis, decision to publish, or preparation of the manuscript.

**Competing interests:** The authors have declared that no competing interests exist.

## Author summary

The collateral blood supply in the cerebral arterial network that covers the brain cortex has an essential role in the prognosis of acute ischemic stroke caused by large vessel occlusion. However, these physiological functions remain poorly understood because of the structural complexities of cerebral arterial anastomoses (the source of collateral blood supply), which have multiscale properties (diameters 50–400 µm). Thus, we aimed to estimate the potential function of multiscale anastomoses in collateral blood supply during acute ischemic stroke using computational modeling. A sparse intermediate scale of anastomoses ($O[10^2]$ µm in diameter) and a dense pial arterial network ($O[10^1]$ µm in diameter) were constructed in the morphologically validated arterial network model (whole-brain scale); blood flow distributions in this arterial network were then computed using numerous patterns of intermediate anastomoses in the MCA occlusion. Obtained results successfully demonstrate that sparse numbers of intermediate anastomoses primarily provide collateral blood supply toward upstream and downstream regions and provide one-order-higher flow recoveries to the occlusion region compared with simulations with a pial network only. We hope that our findings stimulate further clinical and anatomical measurements of multiscale cerebral anastomoses.

## Introduction

The cerebrovascular system is the only source of blood supply to the brain, which has a high metabolic rate and uses 25% of the body's total oxygen consumption from blood flow [1]. To ensure a sustainable blood supply, the cerebrovascular system has a network structure with redundant connections (anastomoses) not only in the vascular bed in the cerebral cortex, but also on cortical surfaces, for robustness against cerebral infarctions. Although there have been numerous studies of cerebrovascular anastomoses over more than a century, their morphologies and functions remain under debate as summarized in [2,3]. Recent medical imaging studies indicate that collateral blood supply by cerebral arterial anastomoses has an essential role in the prognosis of acute ischemic stroke caused by large vessel occlusion [4–6]; their clinical importance is now widely recognized [7–9]. Nevertheless, relationships between the morphological properties of anastomoses and the extent of collateral blood supply remain poorly understood because of limits to both knowledge and measurements.

Cerebral arterial anastomoses have multiscale properties and can be classified by size into three tiers [10]. Tier 1 refers to the anterior and posterior communicating arteries that constitute the circle of Willis, located at the base of the brain. Each anastomose connects the left–right anterior cerebral arteries (ACAs) and the middle cerebral artery (MCA)–posterior cerebral artery (PCA); this is widely accepted to provide system redundancies [11]. Tier 2 refers to the intermediate anastomoses between branches of the three main arteries (ACA, MCA, and PCA). Vander Eecken and Adams [12] examined 10 human cadavers and reported sparse (approximately 10 per whole brain) intermediate anastomoses with a diameter of 200–610 µm between the ACA–MCA and PCA–MCA branches. Subsequent preclinical research has reported that the number of these anastomoses is influenced by genetic factors [13–15]. Tier 3 refers to the local loops formed by pial anastomoses within the same branch. In the first quantitative measurements using 25 human cadavers, Duvernoy et al. [16] reported numerous anastomoses with a diameter of 25–90 µm in pial arterioles on the cerebral cortex. More recently, Blinder et al. [10] conducted a comprehensive study of the network topology of rodent pial arterioles and revealed the characteristic topology of the arterial network, such as

redundancies (network topology is ideally a hexahedral structure) and hierarchical properties (the number of closed loops exponentially decreases with increasing loop area). These hierarchical structures have also been reported in a human cadaver study; the number of anastomoses decreases sharply with increasing anastomoses diameters (ranging from 20–2,000 μm) [17]. Functionally, Blinder et al. [10] suggested that ischemic outcomes following MCA occlusion [18,19] are influenced by intermediate anastomoses (Tier 2), whereas the local rerouting of blood to preserve flow in the face of both local obstructions and a global decrease in perfusion [20] are provided by local loops on the pial arterial network (Tier 3). However, these complementary functions of multiscale anastomoses have received little attention in clinical studies; Tier 2 and 3 anastomoses are thus typically equated as secondary collateral pathways [7].

To clarify the extent of collateral blood supply by multiscale anastomoses, especially in cerebral infarction, computational simulation based on morphologically validated arterial structures on a whole-brain scale is considered a reasonable approach. There are several computational models that represent a whole-brain-scale arterial structure [21–23] by mathematical optimal design based on Murray's minimum energy principle [24]. However, these models assume arterial structures as binary trees without a closed loop, except for the circle of Willis. To the authors' knowledge, a computational model of arterial structures with anastomoses has been proposed by only one research group [25,26]. This group simplified the anastomose structures as a one-layered pial arterial network (Tier 3) consisting of an idealized hexahedral network structure with a diameter of 400 μm (one order higher than those of known pial arterioles [16]); this simplification may overestimate the effects of pial anastomoses on collateral supply. Thus, computational arterial models that consider multiscale arterial anastomoses remain lacking, and little is known about the potential functions of arterial anastomoses for collateral blood supply in cerebral infarction, which is of major clinical interest.

In the present study, we aimed to develop a computational model of multiscale arterial anastomoses in a whole-brain-scale cerebral arterial network, and to then investigate the potential function of anastomoses as resultant collateral blood supply. We hypothesized that intermediate anastomoses (Tier 2) primarily provide collateral blood supply in response to cerebral infarction. To obtain insights into this hypothesis, we developed a multiscale arterial anastomosis model in the whole-brain-scale arterial network developed in our previous study [21] and performed a blood flow analysis in the arterial network to consider both normal and MCA occlusion states.

## Methods

### Whole-brain-scale arterial network model

A cerebral arterial network covering a whole brain cortex was constructed using the method from our previous study [21]; it combined the image-based geometry of the human brain cortex and large cerebral arteries and a mathematical vascular generation algorithm. Here, we briefly summarize the computational framework of the arterial network construction for completeness. Note that all elements needed to construct the arterial network model [21] are publicly available (https://zenodo.org/record/3707179#.Y1YLjnbP1D8).

A cerebral arterial network from the circle of Willis to the pial arterioles was constructed using a hybrid modeling approach that determined these networks by combining large arteries extracted from medical images and a mathematical vascular generation algorithm (multilevel region-confined algorithm) (Fig 1). Here, the arterial structure was represented as a network with sets of nodes $\mathcal{V} = \{|v_m|m = 1, 2, \ldots, |\mathcal{V}|\}$ and edges $\mathcal{E} = \{e_n|n = 1, 2, \ldots, |\mathcal{E}|\}$. In preparation, the brain cortex and large arteries were segmented from medical images (Fig 1(A))

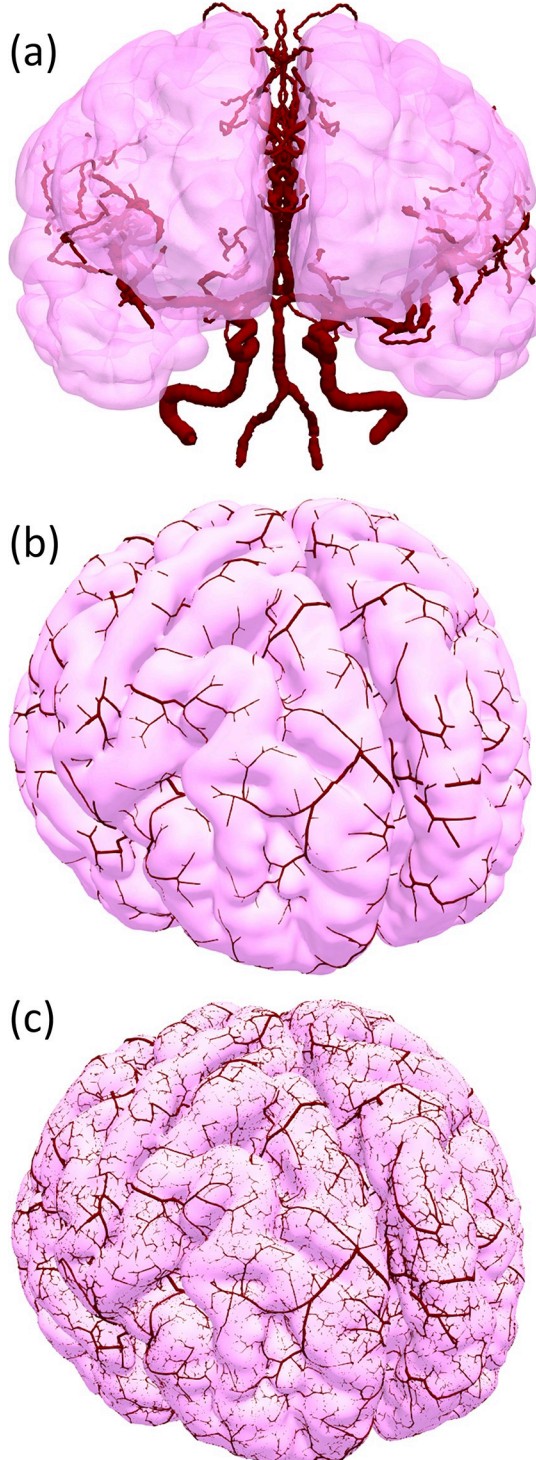

**Fig 1.** Multiscale arterial structure generated using the method from [21]: large arteries and brain surface data were extracted from medical images (a), coarse arteries covered the whole cerebral cortex ($N_1 = 2000$) (b), and fine arteries were distributed in local regions ($N_2 = 37$) (c).

and discretized as closed triangle surfaces, and the network formed arterial centerlines. In the multilevel region-confined algorithm, the vascular network was constructed as a binary tree structure in two steps to express its hierarchical structure. In the first step, a coarse arterial network was constructed from each end node of the large vessels (Fig 1(B)). Each terminal node of the coarse arteries is assigned at each subregion $\Omega_i^c (i = 1, \ldots, N_1)$ of the brain cortical surface, where $N_1$ is the number of coarse arterial subregions. The tree structure was developed based on the minimum energy principle using the constrained constructive optimization algorithm [27] with reasonable simplifications. Here, the total flow rate $Q$ was equally distributed on the brain cortex, and the flow rate $q_i$ at the arterial terminal $i$ was determined by $q_i^c = Q\frac{|\Omega_i^c|}{|\Omega|}$, where $|\Omega_i^c|$ and $|\Omega|$ were the area of the region $\Omega_i^c$ and the whole cerebral cortex, respectively. The fine arterial network was constructed from terminals of coarse arteries in $\Omega_i^c$ independently (Fig 1 (C)). The region $\Omega_i^c$ was further divided into $N_2$ sub-regions $\Omega_i^f$ and one terminal node of the coarse arterial structure was assigned at each $\Omega_i^f$. Here, the arterial edge was divided by 11 nodes to project the arterial edges onto the brain cortical surface. Note that the flow rate $q_i^c$ was only used in the process of the coarse arterial network modeling process and was not applied in the subsequent blood flow simulation.

## Modeling of arterial anastomoses

Fig 2(A) summarizes the three tiers of cerebral arterial anastomoses. Because the circle of Willis (Tier 1) is involved in a subject-specific arterial network extracted from medical images (Fig 1(A)), we modeled intermediate anastomoses (Tier 2) connecting central arterial trees with diameters of $O(10^2)$ μm and pial anastomoses (Tier 3) covering brain cortical surfaces with diameters of $O(10^1)$ μm, as follows.

In the construction of the intermediate anastomoses, boundaries between territories of the RMCA and adjacent main cerebral artery (RACA and RPCA) were segmented by coarse arterial terminals, and all terminal pairs of nearest neighbors between these territories (total number of $N$) were then selected as potential candidates to assign the intermediate anastomoses (Fig 2(B)). Because of the sparse properties of intermediate anastomoses [12], we adopted $n$ numbers of pairs from $N$ numbers of potential candidates to assign intermediate anastomoses. Diameters of the anastomoses were set to an average of connecting arteries.

In the pial arterial network construction, the network structures were modeled as ideal hexahedral loops by the idea of [25]. The hexahedral loops were constructed as the dual graph of surface triangle edges (Fig 2(C)) where the number of nodes was 1,124,504, which was set based on the surface densities of penetrating arteries (8.7 numbers/mm$^2$) [28]. The diameter of the pial arterial network was set to 40 μm based on the known diameters of pial arterial networks and penetrating arteries [16]. Terminals of the fine arterial structure were connected to the nearest nodes of the pial arterial networks. For the subsequent blood flow simulation, we set penetrating arteries with a diameter of 40 μm on each node of the pial arterial networks; the length was set from the edge length of each surface triangle element.

## Blood flow simulation

Blood flow through the arterial network was modeled as a steady zero-dimensional Newtonian flow with a viscosity $\mu$ of 3.5×10$^{-3}$ Pa·s. Poiseuille flow was assumed in each artery; thus, the flow rate $q_{ij}$ in an edge $ij$ with the node $i$ and $j$ was given by

$$q_{ij} = -k_{ij}\left(p_i - p_j\right), k_{ij} = \frac{\pi d_{ij}^4}{128\mu L_{ij}}, \tag{1}$$

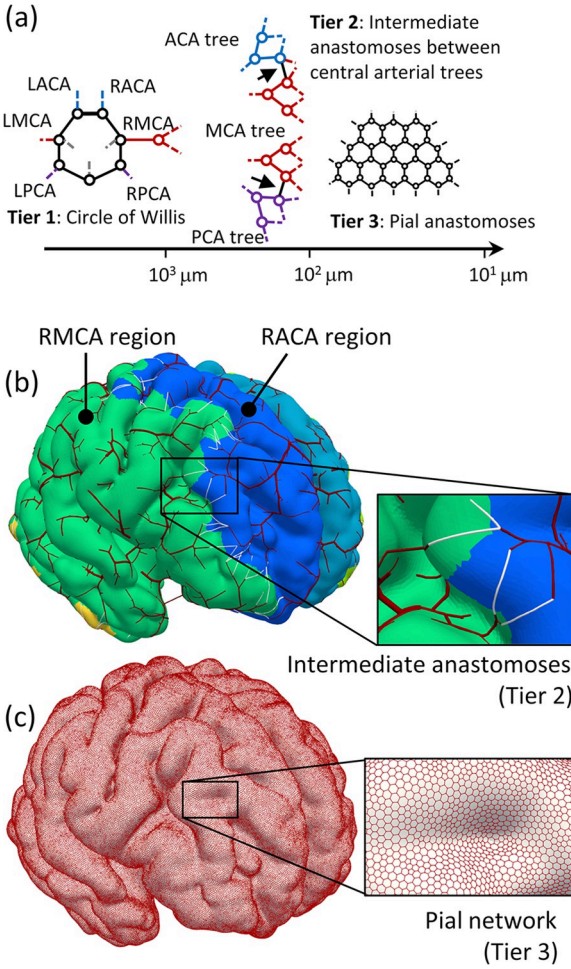

**Fig 2.** Multiscale anastomoses construction. (a) Schematic of multiscale cerebral arterial anastomoses: the main cerebral arteries (anterior cerebral artery [ACA], middle cerebral artery [MCA], and posterior cerebral artery [PCA]) constitute anastomoses (circle of Willis) on the base of the brain (Tier 1); branching patterns of the main cerebral arteries had binary tree structures while intermediate anastomoses were generated between adjacent cerebral arterial trees with an order of $10^2$ μm in diameter (Tier 2); and the number of anastomoses exponentially increased with increasing branches, and a dense pial arterial network structure with an order of $10^1$ μm in diameter was constructed (Tier 3). (b) Computational modeling of the intermediate anastomoses (Tier 2) was set on terminals of adjacent arterial branches (white lines). (c) The pial arterial network (Tier 3) was modeled as hexahedral loop structures, as in [25].

where $p_i$ and $p_j$ are the pressure at the node $i$ and $j$, $d_{ij}$ is the edge diameter, $L_{ij}$ is the edge length, and $k_{ij}$ is the constant determined from the Poiseuille flow assumption. To satisfy the mass conservation at the branching point, the mass-balance equation at the node $i$ was given by

$$\sum_{j}^{n_i} k_{ij}(p_i - p_j) = s_i, \tag{2}$$

Where $n_i$ is the number of nodes connecting to node $i$, and $s_i$ is the source term and is set to zero except for the boundaries. The linear algebraic equation about the pressure was constructed by assembling Eq [2] of all nodes in a whole arterial network. Boundary conditions were as follows: constant pressure of 100 mmHg was set at the inlet nodes of the right and left internal carotid arteries, and right and left vertebral arteries, and a terminal resistance model was assigned at the terminals of pial arteries. As the terminal resistance model, the outlet

pressure $p_i$ at node $i$ was given by

$$p_i = p_0 + Rq_i, \tag{3}$$

where $R$ was the terminal resistance and $p_0$ was the terminal pressure in the vascular bed, respectively. To set $R$, we assumed that the pressure drop in the brain cortex $p_i$–$p_0$ was 65 mmHg [29]; the flow rate $q_i$ at each terminal was given by the areal ratio of the total flow rate of 715 mL/min [30]. Finally, the resultant linear algebraic equation was solved by the algebraic multigrid-preconditioned generalized minimal residual method implemented in the open-source library HYPRE (www.llnl.gov/casc/hypre/).

## Calculation conditions

**MCA occlusion and extent of flow recoveries.** Using the aforementioned computational model, we evaluated the effects of intermediate anastomoses on blood flow distribution under two conditions: normal and right MCA (RMCA) occlusion. For the RMCA occlusion, the diameter of the RMCA arterial edge at the circle of Willis was decreased 99% to represent near-complete occlusion. To evaluate flow recovery to the occluded region, the flow recovery ratio in the occluded region was defined as

$$\text{Flow recovery ratio} = \frac{q_{\text{occlusion}}}{q_{\text{normal}}}, \tag{4}$$

where $q_{\text{normal}}$ and $q_{\text{occlusion}}$ were the flow rates in the normal and occlusion conditions, respectively.

**Effects of hierarchical structure ($N_1$ and $N_2$).** In the arterial structure construction, to consider the effects of the relative position of the intermediate anastomoses, we set three patterns of $N_1$ and $N_2$ (the number of coarse and fine arterial subregions): $(N_1, N_2) = (1000, 74)$, $(2000, 37)$, and $(4000, 19)$. The numbers of resultant arterial edges were approximately $1.5 \times 10^5$ in all cases, and the numbers of candidates of intermediate anastomoses $N$ were 44 at $(N_1, N_2) = (1000, 74)$, 56 at $(N_1, N_2) = (2000, 37)$, and 102 at $(N_1, N_2) = (4000, 19)$. Fig 3 shows a box plot of the terminal edge diameter of each coarse and fine artery. This arterial structure expressed the following multiscale properties: coarse arteries with a diameter of $O(10^2)$ μm broadly covered the brain, and fine arteries were locally distributed in the brain cortex with diameters of $O(10^1)$ μm. The coarse arterial diameters gradually decreased with increasing $N_1$, whereas fine arterial diameters remained within the same ranges; this ensured that these three networks had equivalent branch structure (except for the intermediate anastomoses).

**Effects of numbers of intermediate anastomoses (n).** The effects of the number of intermediate anastomoses on collateral blood supply were evaluated by changing the numbers of intermediate anastomoses. Because there are two ways to assign the anastomosis or not at each terminal pair selected as a potential candidate (Fig 2(B)), the total number of combinations of the assignment of the intermediate anastomoses from N pairs are very large ($2^N$). Therefore, we developed an iterative process to reduce the total number of computations by tracking just the upper limits of the effects of anastomoses, as follows. First, one candidate was selected from all $n \leq N$ candidates, and blood flow distribution was computed in cases of RMCA occlusion in all $n$ cases. The candidate with the maximum total flow recovery ratio in the occlusion was adopted as the anastomosis. The aforementioned three processes iteratively proceeded from $n = 1$ until $n$ became zero. This procedure was able to markedly reduce the total number of the sets for computation to $N(N+1)/2$. The total numbers of computations were 990, 1596, and 5253 in the cases with $(N_1, N_2) = (1000, 74)$, $(2000, 37)$, and $(4000, 19)$, respectively.

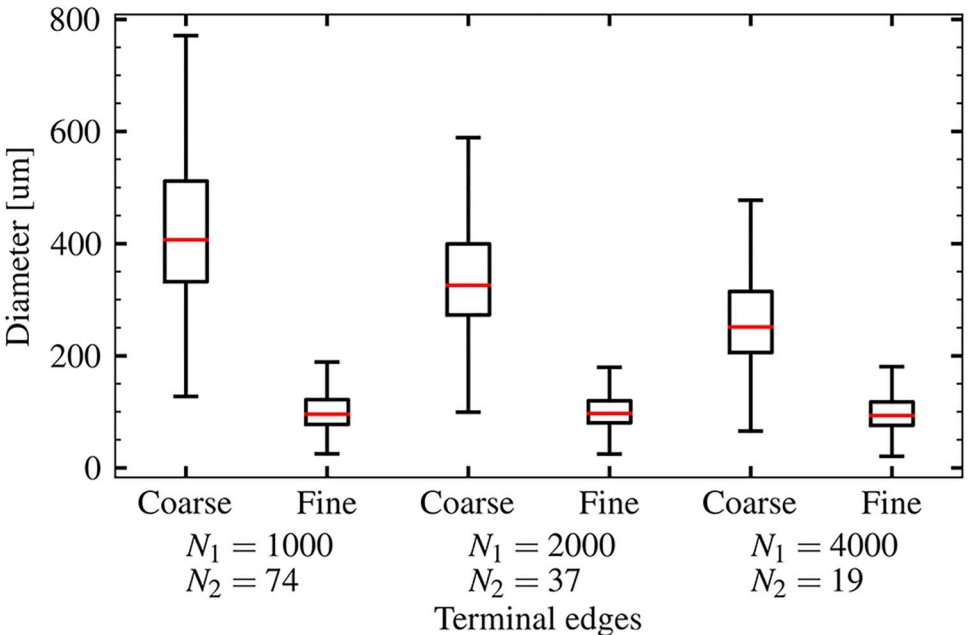

**Fig 3. Box plot of the terminal edge diameter of each coarse and fine arterial structure of three patterns at the number of coarse and fine arterial subregions ($N_1$ and $N_2$) = (1000, 74), (2000, 37), and (4000, 19).** In all three patterns, diameter of the coarse arteries distributed in whole brain surface was in the range of $O(10^2)$ μm whereas that of the fine arteries locally distributed in the brain cortex was in the range of $O(10^1)$ μm.

## Results

### Normal conditions

To confirm the consistencies of the developed computational model with existing physiological measurements, blood flow distribution in the main cerebral arterial branch was evaluated under normal conditions. Fig 4 shows the blood flow rates in each arterial territory in three cases of ($N_1$, $N_2$) = (1000, 74), (2000, 37), and (4000, 19), both without intermediate anastomoses and with the maximum number of intermediate anastomoses. Relative flow distributions were the same in all cases, and differences caused by adding intermediate anastomoses were within 1%. Furthermore, the blood flow distribution was similar to existing measurement data [30].

### MCA occlusion simulations

The blood flow distributions under RMCA occlusion conditions and the collateral blood supply by pial networks without intermediate anastomoses were evaluated using the flow recovery ratio defined as Eq [4] in each main arterial territory in the case of ($N_1$, $N_2$) = (2000, 37) as a representative case, in which the terminal diameters of the coarse arteries were relatively close to the range of the intermediate anastomoses (210–600 μm [12]). Fig 5(A) shows blood flow distribution in each main cerebral arterial territory. The total flow rate in the RMCA region was markedly reduced by occlusion and remained at approximately 3% with the collateral flow of pial anastomoses. In contrast, other arterial territories retained comparable flow rates; differences from those at normal conditions were within 2%. To observe the effects of pial anastomoses on flow recovery, Fig 5(B) shows the spatial distributions of the flow recovery ratio in the RMCA region. Note that the flow recovery ratio was computed in each subregion to which each terminal of the pial arteries was assigned, unless otherwise noted. Pial anastomoses locally

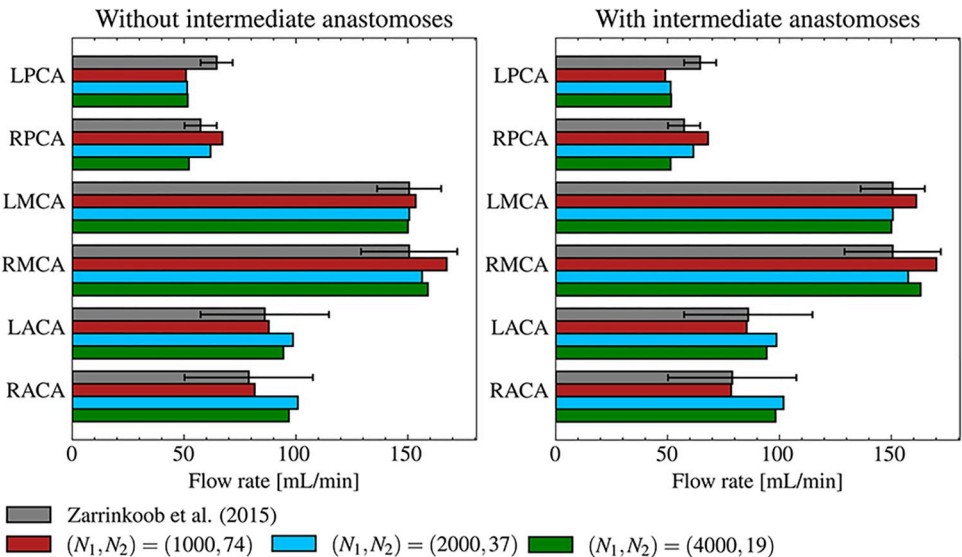

**Fig 4.** Total blood flow rates in each main cerebral arterial territory (left anterior cerebral artery [LACA], right anterior cerebral artery [RACA], left middle cerebral artery [LMCA], right middle cerebral artery [RMCA], left posterior cerebral artery [LPCA], and right posterior cerebral artery [RPCA]) in the case of $(N_1, N_2)$ = (1000, 74) (red boxes), (2000, 37) (blue boxes), and (4000, 19) (green boxes) without intermediate anastomoses (left) and with the maximum number of intermediate anastomoses (right). Gray boxes show the measurement data from [30].

rerouted the blood flow distribution around the boundaries of arterial territories, while the flow recovery ratio sharply decreased from the boundaries and reached less than 10%. The areal fraction of the flow recovery ratio in the RMCA region is shown in Fig 5(C). Flow recovery was less than 5% in the 95% RMCA region.

Fig 6(A) shows both the spatial distributions of the flow recovery ratio in the RMCA region and the flow rate through covering arteries in which the number of intermediate anastomoses ($n$) = 1 in the case of $(N_1, N_2)$ = (2000, 37), which was used as a representative case. The intermediate anastomosis rerouted the blood flow in the RMCA region in both upstream and downstream directions; at most, 40% flow rate was compensated in the downstream branch. When the spatial distributions of the flow recovery ratio with $n$ = 1, 5, and 10 (Fig 6(B)) were considered, the flow recovery ratio was locally higher than 10% at $n$ = 1, whereas a larger region with a flow recovery ratio > 20% occurred at $n$ = 5. There was regional expansion of the high flow recovery ratio with increasing numbers of anastomoses. Fig 6(C) shows the areal fraction of the flow recovery ratio in the RMCA region. Flow recoveries primarily ranged from 5% to 10% in almost all regions (>90%) at $n$ = 1; the flow recovery ratio globally increased with increasing numbers of anastomoses, and was globally higher than 10% in almost all regions (>99%) at $n$ = 5. Furthermore, relatively large variations of flow recovery ratios were noted, and regions in which flow recoveries were higher than 30% reached 2%. These tendencies were also observed in the case of $n$ = 10; the flow recoveries were mainly distributed at more than 15%, with larger variations.

Fig 7 shows the total flow recovery in the RMCA region in relation to the number of intermediate anastomoses. Here, the results of three cases with different $(N_1, N_2)$ were plotted to confirm the variation caused by the relative positions of intermediate anastomoses. In all cases, the flow recovery ratio rapidly increased from 0 to 10 anastomoses and reached 15%–30%. It moderately increased with increasing numbers of anastomoses and asymptotically approached 35%–45%. The increasing degree of flow recovery ratio was higher when the anastomoses were set more upstream. Variation caused by the selection of anastomoses from

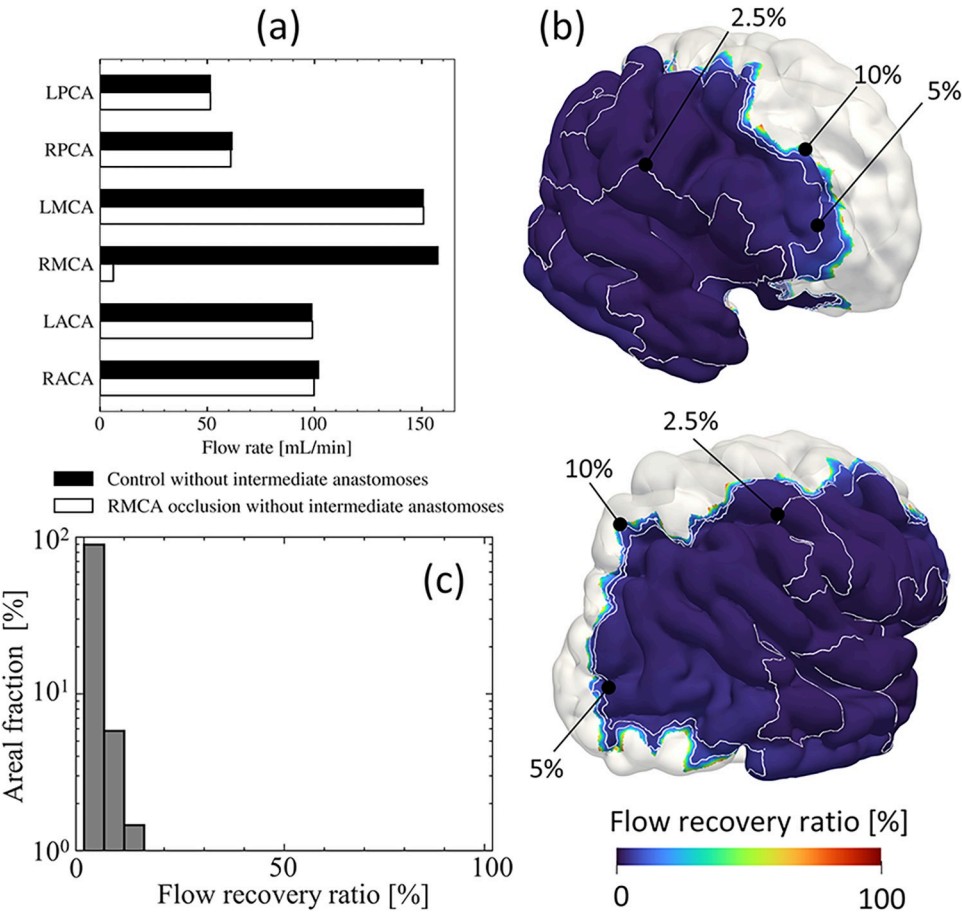

**Fig 5. Blood flow distribution in a case without intermediate anastomoses.** (a) Same as Fig 4, but in normal (black boxes) and RMCA occlusion (white boxes) conditions without intermediate anastomoses. (b) Spatial distributions of the flow recovery ratio in the case of $(N_1, N_2) = (2000, 37)$ without intermediate anastomoses (anterior view [top] and posterior view [bottom]). (c) Volume fraction of the flow recovery ratio on the RMCA (occlusion) region. The flow recovery ratio was computed in each subregion to which each terminal of the pial arteries was assigned, unless otherwise noted.

possible candidates was within 4% at $n = 1$ in all cases, and gradually decreased with increasing numbers of anastomoses.

## Discussion

In the present study, we developed a computational model of multiscale anastomoses in cerebral arterial networks and evaluated the collateral blood supply by anastomoses in MCA occlusion. The morphologically validated cerebral arterial network was constructed using the methods from [21]; intermediate anastomoses with diameters of $O(10^2)$ μm and pial arterial networks with diameters of $O(10^1)$ μm were represented based on current anatomical understanding. The blood flow simulation under normal conditions successfully demonstrated that the flow rate distributions in each main cerebral artery were in good agreement with clinical measurements [30] (Fig 4). This finding demonstrates the capabilities of the developed computational modeling for expressing physiologically reasonable cerebral blood perfusion. Using the aforementioned computational model, *in silico* studies with possible patterns of intermediate anastomoses provided the following insights from clinical and computational viewpoints.

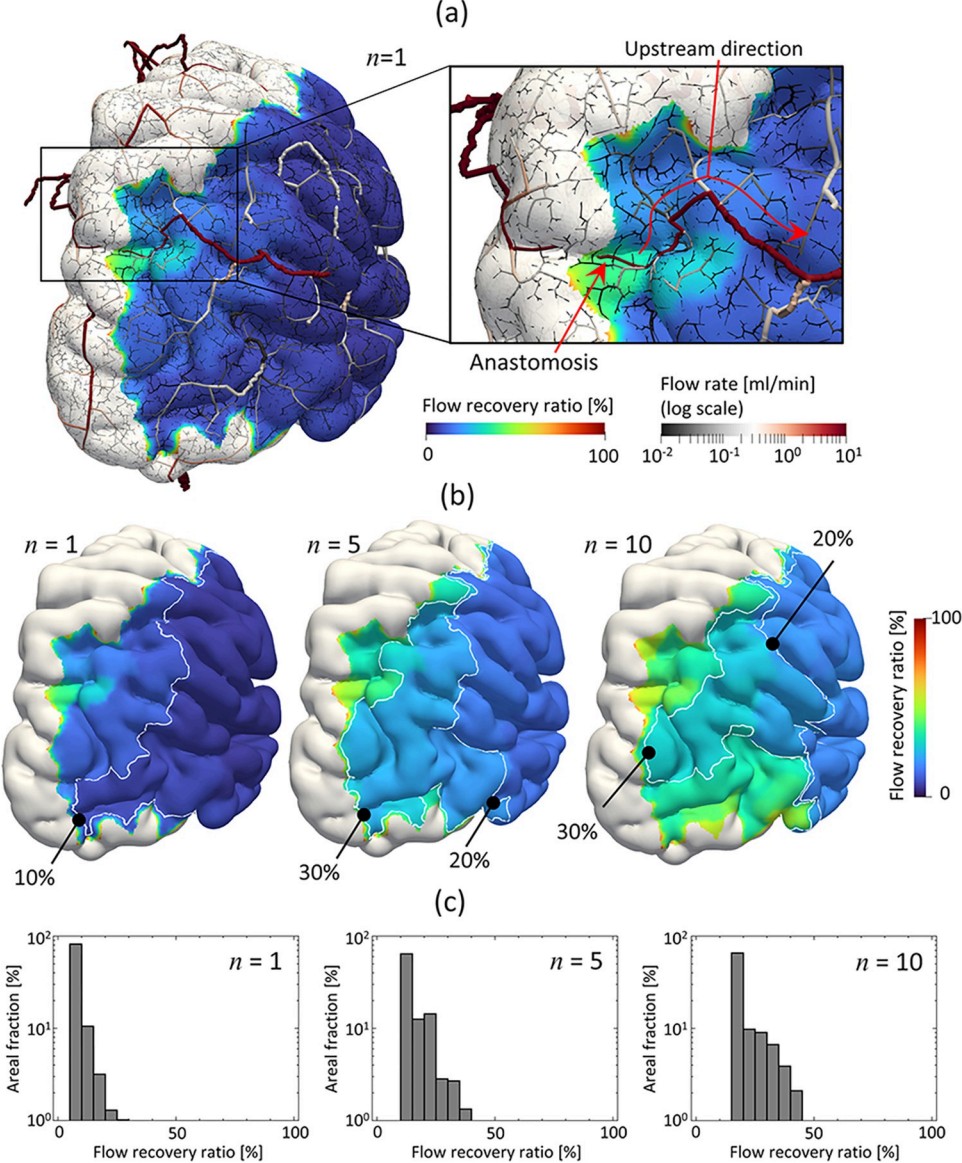

**Fig 6. Blood flow distribution in cases with intermediate anastomoses.** (a) Spatial distribution of the flow recovery ratio in the RMCA domain and flow rates in arteries (pial arteries are not visualized for simplicity) with the number of intermediate anastomoses (n) = 1, and spatial distributions of the flow recovery ratio (b) and its areal fraction (c) at n = 1, 5, and 10 in the case of $(N_1, N_2)$ = (2000, 37).

From a physical viewpoint (e.g., [2]), higher tiers of arterial anastomoses are expected to have higher capacities of blood supply because of larger pressure drops and lower hydraulic resistances. However, the morphological complexities of cerebral arterial networks have made it difficult to estimate the collateral blood supply capabilities of sparse intermediate anastomoses (Tier 2) on a whole-brain scale, and multi-tiers of anastomoses (except for the circle of Willis) have typically been equated as secondary collateral pathways in clinical studies [7]. The present findings indicate that intermediate anastomoses can function as collateral pathways during main cerebral arterial occlusion, even in small numbers. Although the flow recovery was approximately 3% by pial arterial networks (Tier 3) only (Fig 5), even small numbers (≤10) of intermediate anastomoses provided one-order-higher flow recoveries to sufficiently

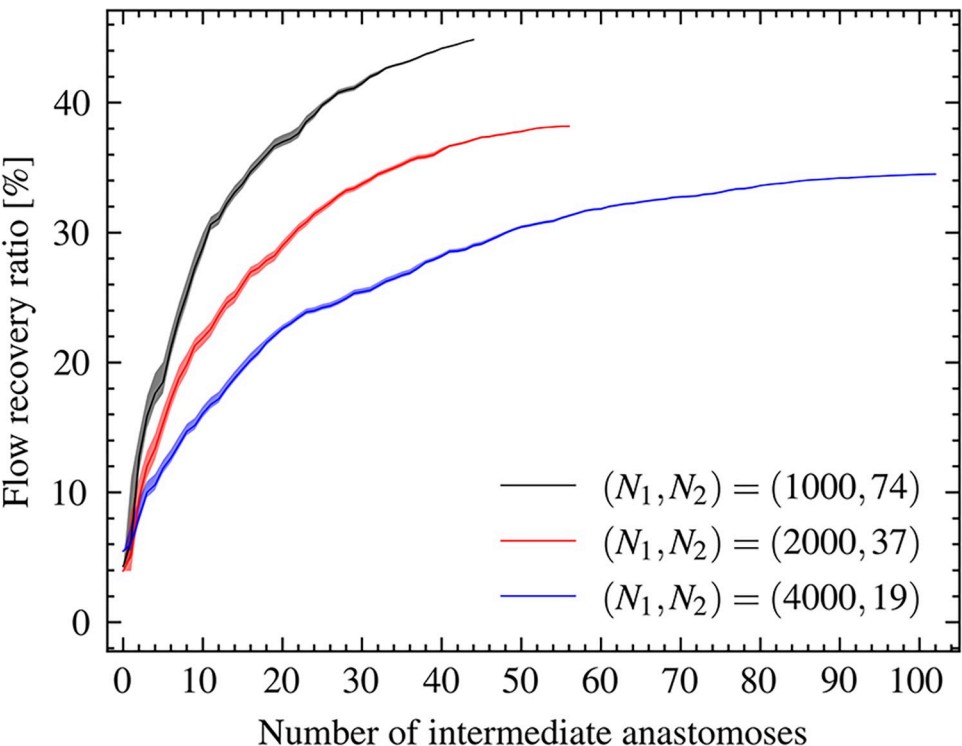

**Fig 7. Total flow recovery ratio in the occluded region (right middle cerebral arterial [RMCA] region) in relation to the number of intermediate anastomoses in the cases of $(N_1, N_2)$ = (1000, 74), (2000, 37), and (4000, 18).** Line widths show the variations in the selection of anastomoses from possible candidates (noted in the *Calculation conditions* section).

reroute blood flow (Fig 7), which is consistent with anatomical evidence [12,17]. These findings indicate that sparse intermediate anastomoses are capable of providing robust cerebral blood perfusion in acute infarction.

Furthermore, the current results revealed two characteristics of collateral blood supply by intermediate anastomoses. First, the selection of intermediate anastomoses from possible candidates of similar size had relatively small effects on total flow recoveries in the occluded region (Fig 7). These findings indicate sufficient collateral blood supply in the upstream direction for global flow recoveries, rather than for local flow recoveries in downstream directions. This feature is consistent with clinical observations that the collateral blood supply in cerebral infarction is visible in a wide range of angiographic images with limited resolutions [8]. Second, a moderate localization of high flow recovery area (20%–50%) was identified around anastomoses and increased with increasing numbers of anastomoses (Fig 6). Regions with moderate blood recoveries by collateral blood supply in the occluded region (25%–50% [31]) are commonly understood to be ischemic penumbra [32], where tissue is viable but functionally depressed because of inadequate perfusion [33]. Because these regions are associated with the possible therapeutic benefit of early blood flow restoration [33], the extent of these formations is of much clinical interest [34]. From this clinical perspective, our findings indicate that intermediate anastomoses are capable of forming ischemic penumbra even in small numbers; this may be a key factor for influencing the extent of ischemic penumbra and subsequent prognosis after cerebral infarction.

From a computational viewpoint, the present results under normal conditions provided further model validation and insights. Under normal conditions, the rerouting functions of

intermediate anastomoses on blood flow supply to the whole brain cortex were minor, regardless of the relative hierarchies of anastomoses (Fig 4). These relatively low flow rates under normal conditions can be understood as small pressure drops in anastomoses, which is consistent with conventional understanding [12]. This consistency supports the assumption adopted in the arterial model [21] of the uniformity in size of each main arterial branch on the boundaries of each territory, even at a coarse scale of $O(10^2)$ μm, and gives further physiological validation to the developed computational arterial model.

The findings concerning the relatively small collateral supply in normal conditions suggest difficulties in patient-specific measurements of anastomose morphologies. Although time-of-flight imaging using 7-Tesla magnetic resonance imaging can now non-invasively detect arterial structures with diameters of >100 μm [35], a relatively low flow rate causes low signal intensity and makes detection challenging compared with other structures of similar diameters. Furthermore, the numbers of intermediate anastomoses are few, and these structures were even not mentioned in a recent cadaver study of cerebral arterial networks with diameters of >100 μm [36]; these features point to difficulties in patient-specific morphological understanding and intermediate anastomose measurements. Further research into the patient-specific detection of arterial anastomoses is thus needed.

This study has fifth main limitations. First, although our model was able to investigate the importance of the multiscale structures of cerebral arterial anastomoses, it was unable to provide quantitative values of patient-specific collateral blood supply because of the limited morphological and physiological data of human cerebral arteries and the complexities of neurobiological processes during occlusion [37]. For example, the geometries of the brain surfaces and the large artery used in this study were extracted from different subjects according to data accessibility [21]. Although this inconsistency does not affect the subsequent modeling of the arterial network because only the terminal nodes of the large arteries were used in this process, this limitation loses information regarding personal specificity, as noted in [21]. Furthermore, autoregulatory function in response to MCA occlusion was not considered because of its complexities [38]. Given that these physiological responses can be considered to recover blood supply, the present study might thus underestimate the extent of collateral blood supply. Second, the present study considered cerebral arterial anastomoses only; there are many other kinds of anastomoses—such as those between intracranial and extracranial arteries or cerebral and cerebellar arteries [39]—that may also provide blood flow in response to the occlusion of main cerebral arteries. Although the anatomical and physiological knowledge of such anastomoses is very limited compared with that of arterial anastomoses, further computational modeling may uncover their potential importance. Third, both intermediate anastomoses and pial arterial networks were ideally set in terms of scales and diameters. In particular, the pial arterial network was modeled as an idealized hexahedral network using the method from [25]. Although this idealization is effective for expressing the global morphological characteristics shown in [10] with a simple algorithm, it does not reflect the hierarchical loop structures identified in the original study [10]. The branching patterns of large arteries, including cerebral arteries with a diameter of $>O(10^2)$ μm [36], obey the minimum energy principle [24]; however, this principle does not accurately express the morphological characteristics of microvasculature ($O(10^0)$–$O(10^1)$ μm) in the human cerebral cortex [40]. The pial arterial network on the brain cortex is located between the aforementioned arterial structures as an interface; thus, further consideration of the functional importance of the pial arterial network will extend our conventional understanding of the physiological optimality of multiscale arterial structures. Fourth, the arterial modeling and blood flow simulation has several assumptions. Because the flow profiles in the peripheral arteries are parabolic shape rather than flat as observed in the aorta [41], we assumed the flow profiles in the whole cerebral arteries to be the steady

Poiseuille flow. However, flow unsteadiness and associated arterial deformations may occur in large arteries around the circle of Willis, while non-Newtonian properties may not be negligible in the small arteries. Further consideration of the above multiscale flow profiles in a whole cerebral arterial structure would provide a detailed quantitative understanding of the cerebral hemodynamics. Fifth, this study investigated the effects of Tier 2 and Tier 3 anastomoses while the relative effects of Tier 1 anastomoses (Circle of Willis) were not considered because the tier 1 anastomoses play a minor role in the case of acute ischemic stroke caused by large vessel occlusion at the post-Circle of Wills, which is the main target of this study. However, in cases where abnormalities occur at the pre-Circle of Wills, such as carotid artery stenosis, full-scale anastomoses including Tier 1 to 3 may help to ensure sustained blood supply, as noted in our previous study [11]. Further systematic analyses of the effects of multiscale anastomoses in various abnormal conditions would provide a comprehensive understanding of the sustainability of the cerebral blood supply.

In conclusion, we developed a computational model of multiscale arterial anastomoses in a whole-brain-scale cerebral arterial network and used it to investigate the extent of collateral blood supply by anastomoses during simulated MCA occlusion. In the morphologically validated computational model of the cerebral arterial network, sparse intermediate anastomoses between adjacent cerebral arterial branches and pial arterial structures were constructed; next, collateral blood supplies were evaluated using a zero-dimensional blood flow model. Even in small numbers ($n \leq 10$), intermediate anastomoses primarily worked to reroute both downstream and upstream blood flow, and the extent of locally high blood recoveries around anastomoses were consistent with previous clinical observations. These findings indicate that a relatively small number of intermediate anastomoses are capable of providing collateral blood supply during cerebral occlusion. We hope that our results stimulate further clinical and anatomical measurements of cerebral arterial anastomoses from all three tiers.

## Acknowledgments

We thank Yutaka Amako for technical assistances and discussions. We thank Bronwen Gardner, PhD, from Edanz (https://jp.edanz.com/ac) for editing a draft of this manuscript.

## Author Contributions

**Conceptualization:** Tomohiro Otani, Shigeo Wada.

**Data curation:** Tomohiro Otani, Nozomi Nishimura, Shigeki Yamada, Yoshiyuki Watanabe.

**Formal analysis:** Tomohiro Otani, Nozomi Nishimura.

**Funding acquisition:** Tomohiro Otani, Satoshi Ii, Marie Oshima, Shigeo Wada.

**Investigation:** Tomohiro Otani, Nozomi Nishimura, Hiroshi Yamashita.

**Methodology:** Tomohiro Otani, Nozomi Nishimura, Hiroshi Yamashita, Satoshi Ii, Marie Oshima.

**Project administration:** Shigeo Wada.

**Resources:** Shigeki Yamada, Yoshiyuki Watanabe.

**Software:** Tomohiro Otani, Nozomi Nishimura, Hiroshi Yamashita, Satoshi Ii.

**Supervision:** Tomohiro Otani, Shigeo Wada.

**Validation:** Tomohiro Otani.

**Visualization:** Tomohiro Otani, Nozomi Nishimura.

**Writing – original draft:** Tomohiro Otani, Nozomi Nishimura.

**Writing – review & editing:** Tomohiro Otani, Hiroshi Yamashita, Satoshi Ii, Shigeki Yamada, Yoshiyuki Watanabe, Marie Oshima, Shigeo Wada.

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
