## [Decision Letter · Decision Letter 0]

29 May 2023

Dear Dr Otani,

Thank you very much for submitting your manuscript "Computational modeling of multiscale collateral blood supply in a whole-brain-scale arterial network" for consideration at PLOS Computational Biology.

As with all papers reviewed by the journal, your manuscript was reviewed by members of the editorial board and by several independent reviewers. In light of the reviews (below this email), we would like to invite the resubmission of a significantly-revised version that takes into account the reviewers' comments.

We cannot make any decision about publication until we have seen the revised manuscript and your response to the reviewers' comments. Your revised manuscript is also likely to be sent to reviewers for further evaluation.

Sincerely,

Andrew D. McCulloch, Ph.D.

Academic Editor

PLOS Computational Biology

Jason Haugh

Section Editor

PLOS Computational Biology

Reviewer's Responses to Questions

**Comments to the Authors:**

Reviewer #1: The review is uploaded as an attachment

Reviewer #2: The authors constructed a computational model of the arterial network in a brain that covers multiple scales from a whole brain to a pial microvessel network. They demonstrated how an anastomosis affected cerebral blood flow in normal and occlusion cases. The main result is that sparse intermediate anastomoses at Tier 2 (branches of ACA, MCA and PCA) play a more important role than Tier 3 (pial anastomoses).

The reviewer thinks that the results are very nice, providing new quantitative understanding of cerebral blood flow with highly sophisticated vessel structures involving multi-level anastomoses. His comments below are rather minor to clarify details of the results.

1. Could the authors theoretically or mechanically explain why Tier 2 anastomoses have a higher impact on collateral blood supply? It is also interesting to compare effects of anastomoses between Tier 2 and Tier 1 (the circle of Willis). These investigations may help quantitatively understand relationships in a cerebral vascular network between the blood flow and vessel diameters and numbers at an anastomose. This may be also related to the flow recovery ratio in Fig. 7; a value at the infinite anastomosis number and a gradient may be related to geometries of vessel network.

2. In L94, should “… (20) is provided …” be corrected to “… (20) are provided …” ?

**Have the authors made all data and (if applicable) computational code underlying the findings in their manuscript fully available?**

Reviewer #1: None

Reviewer #2: Yes

PLOS authors have the option to publish the peer review history of their article (what does this mean?). If published, this will include your full peer review and any attached files.

Reviewer #1: No

Reviewer #2: No
---

## [Decision Letter · Decision Letter 1]

21 Aug 2023

Dear Dr Otani,

We are pleased to inform you that your manuscript 'Computational modeling of multiscale collateral blood supply in a whole-brain-scale arterial network' has been provisionally accepted for publication in PLOS Computational Biology.

Best regards,

Andrew D. McCulloch, Ph.D.

Academic Editor

PLOS Computational Biology

Jason Haugh

Section Editor

PLOS Computational Biology

Reviewer's Responses to Questions

**Comments to the Authors:**

Reviewer #1: The review is not uploaded as an attachment

Reviewer #2: I am happy with the responses and the changes.

**Have the authors made all data and (if applicable) computational code underlying the findings in their manuscript fully available?**

Reviewer #1: None

Reviewer #2: Yes

PLOS authors have the option to publish the peer review history of their article (what does this mean?). If published, this will include your full peer review and any attached files.

Reviewer #1: No

Reviewer #2: No

---

## [Editor Report · Acceptance letter]

4 Sep 2023

PCOMPBIOL-D-23-00430R1 

Computational modeling of multiscale collateral blood supply in a whole-brain-scale arterial network

Dear Dr Otani,

I am pleased to inform you that your manuscript has been formally accepted for publication in PLOS Computational Biology. Your manuscript is now with our production department and you will be notified of the publication date in due course.

With kind regards,

Zsofi Zombor
